# Pretreatment of Glioblastoma with Bortezomib Potentiates Natural Killer Cell Cytotoxicity through TRAIL/DR5 Mediated Apoptosis and Prolongs Animal Survival

**DOI:** 10.3390/cancers11070996

**Published:** 2019-07-17

**Authors:** Andrea Gras Navarro, Heidi Espedal, Justin Vareecal Joseph, Laura Trachsel-Moncho, Marzieh Bahador, Bjørn Tore Gjertsen, Einar Klæboe Kristoffersen, Anne Simonsen, Hrvoje Miletic, Per Øyvind Enger, Mohummad Aminur Rahman, Martha Chekenya

**Affiliations:** 1Department of Biomedicine, University of Bergen, Jonas Lies vei 91, 5020 Bergen, Norway; 2Centre for Cancer Cell Reprogramming, Institute of Clinical Medicine, Faculty of Medicine, University of Oslo, 1112 Blindern, 0317 Oslo, Norway; 3Department of Clinical Science, University of Bergen, 5020 Bergen, Norway; 4Department of Internal Medicine, Hematology Section, Haukeland University Hospital, 5021 Bergen, Norway; 5Department of Immunology and Transfusion Medicine, Haukeland University Hospital, 5021 Bergen, Norway; 6Department of Molecular Medicine, Institute of Basic Medical Sciences, Faculty of Medicine, University of Oslo, 1112 Blindern, 0317 Oslo, Norway; 7Department of Pathology, Haukeland University Hospital, 5021 Bergen, Norway

**Keywords:** autophagic flux, bortezomib, glioblastoma, mitochondrial oxidative respiration NK cells

## Abstract

*Background:* Natural killer (NK) cells are potential effectors in anti-cancer immunotherapy; however only a subset potently kills cancer cells. Here, we examined whether pretreatment of glioblastoma (GBM) with the proteasome inhibitor, bortezomib (BTZ), might sensitize tumour cells to NK cell lysis by inducing stress antigens recognized by NK-activating receptors. *Methods:* Combination immunotherapy of NK cells with BTZ was studied in vitro against GBM cells and in a GBM-bearing mouse model. Tumour cells were derived from primary GBMs and NK cells from donors or patients. Flow cytometry was used for viability/cytotoxicity evaluation as well as in vitro and ex vivo phenotyping. We performed a Seahorse assay to assess oxygen consumption rates and mitochondrial function, Luminex ELISA to determine NK cell secretion, protein chemistry and LC–MS/MS to detect BTZ in brain tissue. MRI was used to monitor therapeutic efficacy in mice orthotopically implanted with GBM spheroids. *Results:* NK cells released IFNγ, perforin and granzyme A cytolytic granules upon recognition of stress-ligand expressing GBM cells, disrupted mitochondrial function and killed 24–46% of cells by apoptosis. Pretreatment with BTZ further increased stress-ligands, induced TRAIL-R2 expression and enhanced GBM lysis to 33–76% through augmented IFNγ release (*p* < 0.05). Blocking NKG2D, TRAIL and TRAIL-R2 rescued GBM cells treated with BTZ from NK cells, *p* = 0.01. Adoptively transferred autologous NK-cells persisted in vivo (*p* < 0.05), diminished tumour proliferation and prolonged survival alone (Log Rank_10.19_, *p* = 0.0014, 95%CI 0.252–0.523) or when combined with BTZ (Log Rank_5.25_, *p* = 0.0219, 95%CI 0.295–0.408), or either compared to vehicle controls (median 98 vs. 68 days and 80 vs. 68 days, respectively). BTZ crossed the blood–brain barrier, attenuated proteasomal activity in vivo (*p* < 0.0001; *p* < 0.01 compared to vehicle control or NK cells only, respectively) and diminished tumour angiogenesis to promote survival compared to vehicle-treated controls (Log Rank_6.57_, *p* = 0.0104, 95%CI 0.284–0.424, median 83 vs. 68 days). However, NK ablation with anti-asialo-GM1 abrogated the therapeutic efficacy. *Conclusions:* NK cells alone or in combination with BTZ inhibit tumour growth, but the scheduling of BTZ in vivo requires further investigation to maximize its contribution to the efficacy of the combination regimen.

## 1. Introduction

Glioblastoma (GBM) is the most frequent and malignant brain tumour in adults [1]. Current multimodal treatment consisting of surgery, radiotherapy and temozolomide-chemotherapy extends survival to only 14.6 months [2], emphasising the need for novel, effective therapies. Amongst the possibilities is immunotherapy, which has hailed a new dawn for the treatment of solid tumours [3,4].

Natural killer (NK) cells may be amenable effectors against GBM due to their ability to spontaneously kill tumours, virus-infected or antibody-coated cells without prior sensitization and through the missing-self mechanism [5,6]. They recognize potential target cells through ligation of their killer immunoglobulin-like receptors (KIR) [7] to cognate inhibitory class I human leucocyte antigens (HLA), which are expressed on all nucleated cells [8,9]. Interaction between the inhibitory KIR-receptors with their relevant HLA ligand results in phosphorylation of the immunoreceptor tyrosine-based inhibitory motif (ITIM) by Src family tyrosine kinases [10,11]. Phosphorylation of this motif creates docking sites for phosphatases SHP-1 and SHP-2 [11] that inhibit NK cell activation by dephosphorylating proteins involved in downstream activation signalling. Killing of the encountered target cell is hindered, thus permitting NK cells to selectively kill diseased targets while sparing healthy self-cells [12]. However, when licensed NK cells encounter cells that lack at least one HLA ligand for their inhibitory KIRs (iKIR-HLA ligand mismatch), activating signals dominate and effective killing of the abnormal target is triggered [12,13]. The HLA class I molecules consist of a major histocompatibility complex (MHC)–encoded heavy chain, the β_2_-microglobulin (β_2_M) light chain and an 8–10 peptide subunit [14]. The peptides are generated by the 26S proteasome and their binding to HLA-class I complex is necessary for HLA expression and stability at the surface [14]. Without peptide, HLA class 1 molecules/β_2_M complexes are retained within the endoplasmic reticulum and Golgi apparatus. Proteasome inhibition reduces the amount of peptide available to bind to HLA class I and decreases its presence at the cell surface. NK cells avidly lyse tumour or virus infected cells that lack surface HLA-ligands [5,10,12], as exemplified by their potent killing of K562 cells that lack HLA-class 1. Clinically, NK cells were demonstrated to mediate potent graft versus leukaemia (GvL) effect when NK cells’ inhibitory KIR fail to recognise HLA-class 1 ligands [15,16]. Proteasome inhibition with bortezomib (BTZ; Velcade^®^) has been demonstrated to downregulate HLA-class I molecules at the surface of multiple myeloma cells, rendering them susceptible to NK cell lysis [17]. More recently, BTZ has been reported to reduce surface levels of HLA-E on myeloma cells and sensitize them to NK cells that expressed CD94/NKG2A as their only inhibitory receptor [18].

NK cells additionally express a diverse set of activating receptors, including NKG2D [19], that recognize stress ligands expressed on cancer cells, e.g., major histocompatibility complex (MHC) chain-related antigens (MICA/B) and the UL16-binding proteins (ULBPs) [20,21]. Ultimately, it is the threshold balance between activation versus inhibition signals that determines NK cell cytotoxicity potential and or cytokine production in their effector phase [10,11]. The major cytolytic pathways involve the release of perforin and granzymes, as well as cytokines such as tumour necrosis factor-alpha (TNFα), TRAIL, interferon gamma (IFNγ) and Fas ligand [22] that ultimately induce programmed cell death of the target. BTZ has also been shown to augment tumour-necrosis-factor-mediated cell death [23,24], induce endoplasmatic reticulum (ER)-stress due to accumulation of mis/unfolded proteins in the ER and promote the expression of stress ligands recognized by the NK cell activating receptors NKG2D and DNAM-1 [25,26]. In addition to the unfolded protein response signaling, blocking the 26S proteasome also impacts autophagy-flux, limiting the availability of pro-survival factors and amino acids from the recycled damaged organelles [27,28,29]. Since Luna et al. recently reported that BTZ sensitizes cancer stem cells from multiple malignancies to allogeneic NK cells [26], we sought to examine whether BTZ could also sensitize patient-derived GBM cells to autologous NK cells and determine the mechanisms of effect. It is established that when mature, autologous NK cells that express at least one inhibitory KIR and licenced on cognate HLA-C, -BW4 or -A3, -A11 ligands encounter a tumour cell that lacks HLA-class 1 ligands, the NK cells will potently kill these targets [8,30]. Nevertheless, not all NK cell subsets in patients are potently cytotoxic [30,31,32]. We, therefore, investigated the hypothesis that GBM cells may be sensitized to bulk NK cells by pretreatment with the proteasome inhibitor bortezomib. BTZ is a US Food and Drug Administration- approved inhibitor for the treatment of multiple myeloma and mantle cell lymphoma [33] and is currently under investigation for combination with NK cells in multiple malignancies (NCT00720785). GBM cell killing by allogeneic and autologous NK cells with or without prior treatment with BTZ was assessed in vitro and subsequently in vivo in an orthotopic xenograft model.

## 2. Materials and Methods

Several methods are described in the Appendix A.

### 2.1. Ethics Approvals and Consent to Participate

The Norwegian regional ethical committee approved the study (REK vest 013.09/20879; 2014/588; 2018-71-5), and samples were collected with the informed consent of patients and healthy donors. All animal procedures were performed in accordance with the European Convention for the Protection of Vertebrates Used for Scientific purposes. The study protocols were approved by The Norwegian Animal Research Authority (Oslo, Norway, FOTS ID 9460). Consent for publication: All authors read and approved the final submitted manuscript.

### 2.2. K562, GBM Cell Culture and NK Cell Isolation

P3, 2012-018, BG7, BG8 and BG9 patient-derived tumour lines were established from primary GBM tissue collected during surgical procedures at Haukeland University Hospital and authenticated as described in the Supplementary Methods. The mutational status of Isocitrate dehydrogenase (IDH) gene in the GBM tumours was determined upon diagnosis by a neuropathologist (co-author HM) at the Department of pathology, Haukeland University Hospital. P3, BG7, BG8 and BG9 were reported to be IDH wildtype, while the IDH status of 2012-018 cells was not available. The human chronic myelogenous leukaemia cell line K562 (American Type Culture Collection, Manassas, VA) was cultured in suspension with RPMI 1640 medium supplemented with 10% fetal calf serum (FCS), 5% penicillin-streptomycin, and 1 mM HEPES at 37 °C and 5% CO_2_. GBM cells were propagated as neurospheres in Neural Basal medium (NB, Invitrogen, Hämeenlinna, Finland) supplemented as previously described [34,35].

Peripheral blood mononuclear cells and NK cells were isolated using the human NK Cell Isolation Kit (Myltenyi Biotec, Bergisch Gladbach, Germany) and expanded as previously described [35]. Briefly, purified NK cells were cultured in CellGro GMP Serum-Free Stem Cell Growth Medium (CellGenix, Freiburg, Germany) supplemented with 10% FCS (PAA Laboratories, Pasching, Austria), 500 U/mL IL-2 (R&D Systems, Abingdon, UK), 5 ng/mL IL-15 (R&D Systems) using the recommended NK Cell Expansion/Activation protocol (Myltenyi Biotec) for 14 days. For in vivo experiments, standardized BG7 neurospheres containing 10^4^ cells/spheroid were established by centrifuging cells mixed in 0.5% *v/v* methylcellulose-NB for 2 h at 2250× *g* rpm and 33 °C in Corning^TM^ Polystyrene V shaped-bottom 96-well plates, followed by culture for 1 week.

### 2.3. Proteasome Inhibitor

Bortezomib (BTZ, Velcade^®^, Cat. no 576415, Janssen, Bergen, Norway) was dissolved in 0.9% *v/v* sodium chloride at 40 μM stock aliquots and stored at −80 °C to ensure drug stability.

### 2.4. Treatment Groups In Vitro

GBM cells were assigned to 3 experimental groups and treated as follows: (1) monotherapy of BTZ, (2) monotherapy of NK cells and (3) combination of BTZ+NK cells. In combination treatment in vitro, NK cells were added following BTZ pretreatment and after removal of drug containing medium.

### 2.5. Clonogenic Assay

Cells were seeded at 1000 cells/well in 6-well plates and after 2 h, experimental groups (1) and (3) were treated with either 6.25 or 12.5 nM BTZ for 24 or 48 h. In treatment groups (2) and (3), GBM cells were treated with NK cells at an effector:target (E:T) ratio of 5:1 for 4 h followed by further observation for 14 days. Colonies were stained with crystal violet and counted as previously described [36].

### 2.6. Flow Cytometry, in Vitro Cytotoxicity Assays and Luminex ELISA

For phenotyping and cytotoxicity experiments, 50,000 GBM cells or 100,000 NK cells were seeded in 96-well plates and, after 1 h (to allow attachment of GBMs cells for conditions (1) and (3)), were treated with different concentrations of BTZ for 24 and 48 h at 37 °C and 5% CO_2_. For treatment groups (2) and (3), a 5:1 E:T ratio of NK cells were added after the removal of media containing BTZ, and further co-cultured for 4 h in RPMI 1640 medium (Invitrogen, Carlsbad, CA, USA) supplemented with 10% FCS (PAA Laboratories, Pasching, Austria), 50 U/mL penicillin-streptomycin (Lonza, Basel, Switzerland), and 1 mM HEPES (Invitrogen) at 37 °C and 5% CO_2_. For combination conditions, NK cells activated in culture for 14 days were used. The cells were washed and labelled with carboxyfluorescein succinimidyl ester (CFSE) (Sigma-Aldrich, St. Louis, MO, USA) according to the manufacturer’s instructions. Culture of target GBM cells alone was used as a negative control. For blocking experiments, anti-TRAIL mAb (RIK-2, BioLegend, San Diego, CA, USA), anti-DR5 mAb (DJR2-4, BioLegend), anti-NKG2D (clone 149810, R&D Systems, Minneapolis, MN, USA), or control IgG1 (clone 11711, R&D Systems) were added during the 4 h cytotoxicity assay, with the concentrations recommended by manufacturers. Each treatment condition was plated in 3 replicates and all cytotoxicity assays were repeated in at least 3 independent experiments. After the different treatments, the cells were washed with phosphate buffered saline (PBS) containing 1% FBS and labelled with a Live/Dead Fixable Near-IR Dead Cell Stain Kit (Invitrogen), according to the manufacturer’s protocol. Cells were stained as previously described [35] with fluorescent conjugated primary antibodies (Appendix A), and Fluorescent Minus One (FMO) controls were used for each channel. For the gating strategy involving phenotyping acute dissociated tumours and NK cells from tumours, cells were gated from debris and doublets; singlets and live cells were selected for further analysis. Target cells (GBM or NHA) were identified as CFSE negative (−), and effector cells (NK cells) were identified as CFSE positive (+). For cytotoxicity analyses, the dead target cells were identified as CFSE− and Live/Dead+. Spontaneous death was defined as the proportion of dead cells in the negative control, and this value was subtracted from the proportion of dead target cells co-cultured with effector cells. Data was acquired on a BD LSR Fortessa flow cytometer (BD Biosciences, Trondheim, Norway) and analyzed using FlowJo, version 10 (Tree Star Inc.; Ashland, OR, USA). Supernatants from replicate treatment groups from each independent experiment were pooled and stored at −80 °C for Luminex cytokine analyses, which were performed as previously described [35].

### 2.7. Tandem Fluorescent-Tagged LC3 (mCherry-EGFP-LC3B) Plasmid, Transfections and Gene Transduction

GBM cells (0.25 × 10^5^) or NK cells (0.5 × 10^5^) were seeded on 12 mm cover slips (cat. no. 631-1577, VWR) in 24-well plates. After 2 h, cells were treated with NK cells alone, 6.25 or 12.5 nM BTZ for 24 h or in combination with NK cells at an E:T ratio of 5:1 for 4 h, where NK cells were added after BTZ treatment. N-terminal double-tagged (mCherry-EGFP) LC3B was cloned into the pLVX-Tight-Puro vector (Takara Bio USA, Mountain View, CA, USA) to generate the pLVX-Tight-Puro-mCherry-EGFP-LC3B construct. Plasmids were transfected into 293T cells using BBS/CaCl_2_ to produce lentivirus from the pLVX-Tet-On-Advance (Takara Bio USA) regulator vector and the pLVX-Tight-Puro-mCherry-EGFP-LC3B response vector. For infection, the viral supernatants containing both plasmids were added to the cells in the presence of polybrene (4 µg/mL) and with or without doxycycline (750 ng/mL; Sigma Aldrich). The mixtures were centrifuged at 2225× *g* rpm for 90 min at room temperature. After 24 h, the supernatant was removed and replaced with medium with or without doxycycline. Positive cells were selected for in puromycin (1 µg/mL; Sigma Aldrich), followed by FAC-Sorting using the Sony SH800 (Sony Biotechnology, San Jose, CA, USA).

### 2.8. Autophagic Flux Assay

P3 cells (2.5 × 10^4^) transduced with tandem fluorescently-tagged LC3 (mCherry-EGFP-LC3B) plasmid (described above) were cultured on coverslips in 24-well plates overnight, and then treated according to the different schedules. Chloroquine (CQ), an inhibitor of autophagy was added at a final concentration of 50 μM 24 h before NK treatment. Cells were washed, slides mounted with Prolong Gold DAPI (P36931, Molecular Probes, Eugene, OR, USA) and confocal images acquired on a Leica T SC SP5 3X (Leica, Wetzlar, Germany). The number of LC3B-positive autophagosomes and autolysosomes in merged images was quantified in at least 10 cells from each treatment group.

### 2.9. Seahorse Mitochondrial Stress Test

Cells were evaluated using the Seahorse assay (Agilent Seahorse XFe96 Cell Mito Stress Test Kit, Agilent Technologies, Santa Clara, CA, USA) which assesses key parameters of mitochondrial function by measuring the oxygen consumption rate (OCR) after sequential drug challenge. The concentration of these drugs was optimized for each type of cell following the manufacturer’s instructions. P3 and 2012-018 cells were plated in microplates and treated as described above; NK cells were carefully washed 2 times with 1×PBS before OCR measurements of tumour cells were taken. NK cells were plated at a density of 0.5 × 10^6^ cells/100 µL/well in microplates previously coated with Corning^®^ CellTak^TM^ (cat. no. 354240, Fisher Scientific, Oslo, Norway) following the manufacturer’s instructions, and were left untreated or treated with 6.25 or 12.5 nM of BTZ for 24 and 48 h at 37 °C and 5% CO_2_.

### 2.10. Proteasome Activity Assay

Catalytic activity of the 20S proteasome subunit was measured following the manufacturer’s instructions (APT280, Merck KGaA, Darmstadt, Germany). Briefly, the lysis buffer (50 mM HEPES (pH 7.5), 5 mM EDTA, 150 mM NaCl and 1% Triton X-100) was used to extract the proteasomes from the tissue obtained from the brains of mice (*n* = 3/group) treated with BTZ, NK or in combination. Proteasomes (10 µg) were mixed with 1× Assay Buffer. Fluorogenic peptides LLVY-AMC were mixed into the reaction and incubated at 37 °C for 2 h to assess chymotrypsin-like activity of the 20S proteasome. Fluorescence was measured using a plate reader (Asys UVM 340, Biochrom, Holliston, MA, USA) at 355(excitation)/460(emission). The fluorophore 7-amino-4-methylcoumarin (AMC) and proteasome positive control standard curves were generated by reading a serial dilution of the reconstituted AMC standard and the proteasome positive control included in the kit.

### 2.11. Treatment of GBM-Bearing Mice

NOD/SCID mice (7 weeks; *n* = 36; Taconic) were implanted intracranially with 5 standardised BG7 tumour spheroids as previously described in [35] and briefly recounted above. Nineteen days after implantation, mice were treated under one of the following protocols: (1) 5 µL PBS containing 4.4 g/L glucose (vehicle control: *n* = 8); (2) BTZ monotherapy 0.5 mg/kg, corresponding to 1.3 mg/m^2^, the recommended human equivalent dose (HED), calculated according to federal drug administration (FDA) guidelines [37] (*n* = 6); (3) monotherapy of 0.5 × 10^6^ autologous purified NK cells from the BG7 patient diluted in 5 µL of PBS containing 4.4 g/L glucose (*n* = 7); or (4) combination of BTZ 1.3 mg/m^2^ for 24 h followed by intralesional injection of 0.5 × 10^6^ autologous NK cells (*n* = 8); (5) combination of BTZ 1.3 mg/m^2^ for 24 h followed by intralesional injection of 0.5 × 10^6^ autologous NK cells inhibited with anti-asialo-GM1 antibody (1:10 for 24 h; *n* = 7). PBS and NK cells were administered into the same coordinates as tumour spheroids. BTZ was administered intraperitoneally on days 1, 4 and 8, and animals were weighed daily and monitored for survival. Animals were sacrificed by CO_2_ inhalation and dislocation of the neck when they developed neurological symptoms, such as rotational behaviour, reduced activity, delayed grooming, or loss of bodyweight ≥20%. Brains were subsequently removed and cut in two. One portion was fixed in formalin for H&E and immunohistochemistry staining, and the other was dissociated into single cells as previously described [35].

### 2.12. LC-MS-MS Assay for Detection of Bortezomib in Mouse Brain Tissue

A quantity of 250 µL of acetonitrile (Honeywell, Seelze, Germany) containing bortezomib-d_8_ (Toronto Research Chemicals, Toronto, ON, Canada) as an internal standard, and 250 µL of a 1% aqueous solution of formic acid (Sigma-Aldrich, Steinheim, Germany) were added to cryopreserved mouse brain tissue. The samples were then homogenized in a Tissuelyser ball mill from Qiagen (Crawley, UK), and thereafter, sonicated for 5 min and centrifuged for 10 min at 2 °C. A total of 100 µL of the supernatant was removed and evaporated to dryness under flowing nitrogen. A total of 100 µL of a 1% *v/v* formic acid was used for reconstitution, and the samples were then subjected to liquid–liquid extraction by adding 700 µL methyl tert-butyl ether (Sigma-Aldrich). A quantity of 600 µL of the supernatant was evaporated to dryness and reconstituted in 100 µL of acetonitrile: 0.01% formic acid (50:50, *v/v*) and analysed on an Acquity Ultra Performance liquid chromatography system connected to a Xevo TQ-S tandem mass spectrometer (Waters, Milford, MA, USA). The compounds were separated on a Waters BEH C_18_ column (50*2.1 mm, 1.7 µm particle size), which was developed by gradient elution over 14 min using water: acetonitrile (9:1, *v/v*) as weak mobile phase and acetonitrile as a strong mobile phase. Bortezomib was detected in the positive mode by using the 367.1 > 226.0 transition as a quantifier and the 367.1 > 207.9 as a qualifier. The ratio between the quantifier and qualifier was calculated to check the purity of the chromatographic peaks. Similarly, the 375.2 > 233.9 and 375.2 > 215.1 transitions were used for bortezomib-d_8_. Brain tissue from untreated mice was used for calibration after adding known amounts of bortezomib standard (Toronto Research Chemicals, North York, ON, Canada) to the precipitation solution. Final concentrations of bortezomib were adjusted for the weight of each brain sample, which ranged from 11 mg to 36 mg. Bortezomib was stable in acetonitrile: aqueous formic acid for 48 h at room temperature (99 ±16.8%) and during five freeze–thaw cycles (108 ± 13.3%). They were also stable for one hour after adding the precipitation solution and for 24 h in the autosampler (95 ± 10.8%).

### 2.13. Magnetic Resonance Imaging and Tumour Volume

Magnetic resonance imaging (MRI) was performed as described in the Supplementary Methods.

### 2.14. Immunohistochemistry

Histological staining of formalin-fixed paraffin-embedded (FFPE) brain sections from 4 animals from each treatment group was performed as previously described [35], and the methods are briefly described in the Supplementary Methods. Whole brain images were acquired with the VS120 Virtual Slide Microscope (Olympus Corporation, Tokyo, Japan).

### 2.15. Statistical Analysis

The animal survival data were analysed using the Kaplan–Meier method with the Mantel–Cox log rank test [38]. When comparing more than two groups with one dependent variable, a one-way ANOVA was used. A two-way ANOVA was used to analyse data with two or more dependent variables compared in two or more groups. Bonferroni or Tukey’s post hoc correction for multiple testing was used. *P* < 0.05 was considered to be statistically significant (shown as * *p* < 0.05, ** *p* < 0.01, *** *p* < 0.001, and **** *p* < 0.0001). All experiments were performed in triplicate and repeated in at least 3 independent experiments, and graphs represent the mean ± the standard error of the mean (SEM) of at least 3 independent experiments. All statistical analyses were performed using the GraphPad Prism v6.07 (La Jolla, CA, USA).

## 3. Results

### 3.1. Heterogeneity in NK Cell Cytotoxic Potency against GBM Cells

NK cells from healthy donors and GBM patients were assessed for the ability to lyse GBM cells in vitro. The cytotoxicity of allogeneic NK cells from healthy donors (*n* = 9) varied between 11–40% of GBM cells killed at increasing effector: target ratios in a standard 4 h flow cytometry assay (Figure 1A,C), and 24–46% at an effector:target (E:T) ratio of 5:1 when evaluated in longer-term clonogenic survival assays (Appendix A). As expected, NK cells killed 50% of the susceptible K562 cells that were used as controls at 5:1 E:T ratio (*p* < 0.0001, Figure 1B). We also investigated the ability of patients’ own NK cells to kill their corresponding GBM cells (Figure 1D). Activated autologous NK cells killed 22% of BG7 and BG8 cells, while BG9 tumour cells were highly resistant to lysis (*p* < 0.001, Figure 1D). However, inhibition of allogeneic/autologous NK cell activity with antibodies against asialo-GM1 ganglioside abrogated the lysis of target cells (P3 and BG8), confirming the specificity of lysis by NK (*p* < 0.01, Figure 1E).

NK and tumour cells were examined for expression of molecules involved in NK-mediated lysis. Greater proportions of P3 and 2012-018 cells expressed MICA and ULBPs, whereas BG7 cells expressed MICB and CD112 (Figure 1F,G), which are stress ligands recognized by NKG2D and DNAM-1 NK-cell-activating receptors, respectively. The KIR–HLA ligand interactions were investigated based on genotype, where licenced and potentially functional subsets were identified (Appendix A). NK cell expression of these activating and inhibitory receptors at the protein level was confirmed by flow cytometry (Figure 1H–J). Taken together, both allogeneic and autologous NK cells variably killed GBM cells, where potency was modulated by a relative balance of inhibitory vs. activating receptor–ligand interactions.

### 3.2. NK Cells Secrete Cytotoxic Factors that Impair Mitochondrial Function in GBM Cells

To determine the mechanisms of NK-cell-mediated GBM cell killing, we investigated the levels of cytotoxicity factors released into the supernatant from NK cells co-cultured with tumour cells in vitro. Upon contact with both P3 and 2012-018 GBM cells, NK cells released increased levels of perforin (*p* < 0.01 and *p* < 0.0001, respectively, Figure 1K) and granzyme A (*p* < 0.0001) cytolytic granules compared to NK cells alone (Figure 1K). NK cells in contact with 2012-018 cells also released increased levels of IFNγ compared to NK cells alone (*p* < 0.01, Figure 1K). We next investigated whether treatment with NK cells impacted the mitochondrial function of P3 and 2012-018 GBM cells (Figure 1L). Based on the OCR obtained in the Seahorse assay, maximal respiration and ATP production were significantly reduced in both P3 and 2012-018 GBM cells after treatment with NK cells (*p* < 0.01 and *p* < 0.05, respectively, Figure 1M). The spare respiratory capacity was also reduced in P3 cells under NK cell treatment, revealing a deficiency in coping with the additional energy demand (in response to antimycin A and rotenone) (*p* < 0.05, Figure 1M). In contrast, the response of 2012-018 cells to the additional energy demand remained unaffected (*p* > 0.05, Figure 1M). This result was consistent with their increased resistance to NK cytolysis relative to P3 cells. Taken together, NK cells induced oxidative stress and disrupted mitochondrial function in GBM cells.

### 3.3. Bortezomib Is Cytotoxic Against Patient-Derived GBM Cells In Vitro

To explore the hypothesis that pretreatment of GBM with BTZ might sensitize patient tumours to lysis by autologous NK cells, we first investigated the efficacy of BTZ against P3, 2012-018, and BG7 cells, and determined its half maximal inhibitory concentration (IC_50_) after treatment for 24 or 48 h. BTZ was cytotoxic in a dose- and time-dependent manner with enhanced susceptibility of all cell populations after 48 h of treatment (Appendix A). Normal human astrocytes were more resistant to BTZ than P3 and BG7 but not 2012-018 tumour cells. The doses required to kill 50% of cells were higher when assessed in the 4 h flow-cytometry-based cytotoxicity assay compared to long-term clonogenic assays which yielded more physiologically relevant doses (Appendix A). Thus, we used 6.25 and 12.5 nM doses derived from the clonogenic assays for investigation of GBM sensitization to NK cells following 24 h BTZ treatment.

### 3.4. Combination BTZ+NK Cell Treatment Is More Effective against GBM than Monotherapy In Vitro

NK cells alone showed variable efficacy against the different GBM cells in vitro. However, pretreatment of P3 cells with either dose of BTZ for 24 h decreased cell survival to ~40–50% in the presence of allogeneic NK cells (Figure 2A, *p* < 0.01; and Figure 2B, *p* < 0.05). 2012-018 cells were also more sensitive to the combination treatment, although statistical significance was reached only at the higher dose of BTZ (12.5 nM; *p* < 0.05, Figure 2B). The IC_50_ dose of BTZ for 24 h pretreatment of GBM cells was significantly reduced, 3.3 nM for P3 (*p* < 0.05) and 5.4 nM for 2012-018 (*p* < 0.001), when combined with NK cells at an effector target ratio of 5:1 (Appendix A). BG8 tumour cells were also sensitized to autologous NK cells after BTZ pretreatment, from 20% to 40% of cells lysed at 12.5 nM (*p* < 0.05, Figure 2D).

However, BG7 tumour cells were already more sensitive to low doses of BTZ (7.5 nM) at 24 h than killing by NK cells and thus, were not further sensitized by combination with allogeneic or autologous NK cells (Appendix A, Figure 2A–D). Finally, BG9 cells remained resistant to NK cell cytolysis with or without BTZ pretreatment (Figure 2C,D). Similar results were observed after 48 h of BTZ treatment (Appendix A) and in long-term clonogenic survival assays (Appendix A). Taken together, allogeneic and autologous NK cell cytotoxicity was enhanced in three out of five GBM cell populations after pretreatment with BTZ.

### 3.5. Combination of Bortezomib+NK Cell Treatment Kills GBM Cells by Apoptosis

Having established that NK cell cytotoxicity against GBM cells can be enhanced by combination treatment with BTZ, we next sought to establish the mode of cell death. Annexin V^+^ staining with detection by flow cytometry revealed that NK cell monotherapy, as well as combination BTZ+NK cell treatment, induced apoptotic cell death in P3 (*p* < 0.0001 for all, Figure 2E) and BG7 cells (*p* < 0.05 NK cell monotherapy; *p* < 0.001 6.25 nM and *p* < 0.01 12.5 nM combination BTZ+NK cell treatment, Figure 2E). 2012-018 cells were consistently more resistant to these therapies than P3 or BG7 and did not show evidence of apoptosis (Figure 2E). BTZ monotherapy, however, did not induce cell death through apoptosis in any of the cell populations (Figure 2E).

Changes in markers of apoptosis were consistent with these results as the cleavage of poly ADP-ribose polymerase-1 (PARP-1) and caspase-3 was induced after NK cell treatment, while BTZ monotherapy did not induce PARP-1 or caspase-3 cleavage (Figure 2F). Interestingly, combination BTZ+NK cell treatment led to moderate cleavage of PARP-1 and caspase-3 in P3 cells (Figure 2F). Taken together, NK cells killed GBM cells by apoptosis with corresponding PARP-1 and caspase-3 cleavage.

### 3.6. Bortezomib Pretreatment Enhances Expression of Stress Ligands in GBM Cells

We examined the effect of BTZ pretreatment on the expression of stress ligands in GBM cells using flow cytometry. BTZ pretreatment enhanced baseline expression of MICA (in BG8: 6.25 nM, *P* < 0.001 and 12.5 nM, *P* < 0.01), MICB (in P3: 12.5 nM, *P* < 0.0001; and BG8: 6.25 nM, *P* < 0.01), ULBP1 (in BG7: 6.25 nM, *P* < 0.01, and 12.5 nM, *P* < 0.0001; and BG8: 6.25 and 12.5 nM, both *P* < 0.05) and ULBP3 (in P3: 6.25 and 12.5 nM, both *P* < 0.0001; in BG7: 6.25 nM, *P* < 0.01, and 12.5 nM, *P* < 0.001; and BG8: 6.25 and 12.5 nM, both *P* < 0.001; Figure 2G,I) which are all recognized by the NKG2D-activating receptor on NK cells. BTZ also enhanced expression of TNF-related apoptosis-inducing ligand receptor (TRAIL-R2 or DR5) by ~4–6 times in all cell populations, except for 2012-018 (in P3: 6.25 and 12.5 nM, both *P* < 0.05; in BG7: 6.25 nM, *P* < 0.01, and 12.5 nM, *P* < 0.0001; and BG8: 6.25 nM, *P* < 0.05, and 12.5 nM, *P* < 0.01; Figure 2G,H; and mean of fluorescence intensity (MFI), which enables comparison between intensities of fluorescence relative to untreated control, Appendix A). TRAIL-R2 induces extrinsic apoptosis upon ligation of the receptor by its ligand. These expression levels of TRAIL-R2 were consistent with reduced apoptosis in 2012-018 GBM cells.

### 3.7. Blocking NKG2D, TRAIL and TRAIL-R2 Rescues GBM Cells from NK Lysis

As proof of concept, we sought to determine whether the apoptotic cell death was mediated via extrinsic or intrinsic pathways. We blocked TRAIL-R2 (DR5) on GBM cells, as well as NKG2D and TRAIL on NK cells before and after treatment with 12.5 nM BTZ. NK-cell-mediated lysis of P3 cells without BTZ pretreatment was attenuated when NKG2D and TRAIL-R2 interactions with their cognate ligands were blocked (NK cells only vs. NK+ anti-NKG2D and vs. NK+ anti-TRAIL-R2, both *p* < 0.05, Figure 3A). Blocking TRAIL and isotype control had no significant effect on NK cell lysis of GBM cells that were not treated with BTZ (*p* > 0.05, Figure 3A), indicating that the interaction of NKG2D and stress ligands was the most potent cytotoxicity signal. Intriguingly, blocking NKG2D and TRAIL on NK cells, as well as TRAIL-R2 on GBM cells after treatment with 12.5 nM BTZ, rescued the cells from NK cell lysis more significantly (all *p* < 0.01, Figure 3A,B,D–F), where isotype control had no effect (*p* > 0.05, Figure 3A,C). Taken together, these findings indicate that NK cell treatment alone promoted intrinsic apoptosis mediated via the ligation of stress ligands to the NKG2D-activating receptor. However, since BTZ pretreatment elevated TRAIL-R2 expression by GBM cells, combination treatment with NK cells augmented extrinsic apoptosis. This is supported by the finding that blocking of both TRAIL and TRAILR2, rescued the GBM cells from NK cell lysis.

Mitochondrial dysfunction occurs early during the induction of intrinsic apoptosis [39]. Therefore, mitochondrial function of P3 and 2012-018 GBM cells under BTZ monotherapy or NK+BTZ combination therapy was examined by the OCR in the Seahorse assay (Appendix A, respectively). BTZ alone or in combination with NK cells did not further decrease GBM mitochondrial function compared to NK cell monotherapy (Appendix A), confirming NK cells’ potent role in GBM cell lysis. Furthermore, given that oxidative stress, mitochondrial function and turnover are intricately linked to autophagy [27], we examined the role of BTZ in inducing cell death in GBM cells through abrogation of autophagic flux, as has been previously shown [28]. BTZ interrupted autophagic flux in P3 and BG7 GBM cells regardless of the treatment regimen (Appendix A).

### 3.8. Combination Bortezomib+NK Cell Treatment of GBM Enhances NK Cell Secretion of IFNγ

To determine the mechanism of enhanced GBM killing by BTZ+NK cell combination treatment, we compared levels of cytotoxicity factors released by NK cells into the supernatant in vitro. Treatment of P3 cells with combination BTZ+NK cells led to significantly increased IFNγ secretion by NK cells compared to NK cells alone or NK cells treated with 12.5 nM BTZ (*p* < 0.05, for both, Figure 2J). P3 cells alone did not secrete any IFNγ. Therefore, pretreatment of GBM cells with BTZ augmented the activation of allogeneic NK cells, as indicated by the enhanced IFNγ release.

### 3.9. Bortezomib Promotes Maturation of NK Cells into an Activated Phenotype

In vivo, NK cells would be exposed to BTZ due to the inability to remove the drug before adoptive transfer. Therefore, we investigated whether BTZ might be cytotoxic to activated NK cells. Survival of NK cells decreased with increasing doses of BTZ. Treatment of activated NK cells with 12.5 nM BTZ for 24 h killed 10% more NK cells and induced early apoptosis in 27.3% more cells than 6.25 nM treatment (*p* < 0.001 compared to untreated activated NK cells, Figure 4A–C). However, NK cell viability was significantly reduced after 48 h of treatment compared to 24 h (*p* < 0.0001, Figure 4A). After BTZ treatment for 24 h, viable NK cells also exhibited more mature, activated and cytotoxic CD57^+^CD16^dim^CD69^+^ phenotypes (Figure 4D,E). Since NK cell stimulation and IFNγ production as a result of ligating activating NK receptors requires oxidative phosphorylation [40], we examined the OCR-determined mitochondrial function of NK cells after 6.25 nM and 12.5 nM BTZ treatment for 24 h. There was limited perturbation of the OCR (Figure 4F–H), confirming their relative intact mitochondrial function.

### 3.10. Autologous NK Cells Alone or in Combination with Bortezomib Diminishes Tumour Volume and Prolongs Survival in Tumour Bearing Animals

To investigate the efficacy of combination BTZ+NK cell treatment in vivo, we treated mice 19 days after the implantation of BG7 spheroids. Mice were pretreated with BTZ at 1.3 mg/m^2^ for 24 h prior to a single intratumoural infusion of 0.5 × 10^6^ activated purified autologous NK cells (Figure 5A). In animals treated with vehicle or BTZ monotherapy, tumours developed rapidly as large, diffusely invasive tumours with hypointense cores and ring-enhancement on a T1-weighted magnetic resonance imaging (MRI) with contrast, indicative of angiogenesis and central necrosis (Figure 5B). In contrast, tumour growth was strongly attenuated in combination BTZ+NK cells or NK-cell monotherapy-treated animals (Figure 5B). The size of BTZ-treated tumours was also significantly greater than those treated with NK monotherapy and combination BTZ+NK cells (*p* < 0.01, for both, Figure 5B,C).

To determine the contribution of NK cells to the efficacy of the combination treatment, we ablated NK cell function by introducing anti-asialo-GM1 ganglioside into the treatment. Animals treated with BTZ+anti-GM1+NK cells had very similar disease progression as those treated with BTZ monotherapy, where tumour volumes and survival showed no difference (*p* > 0.05, Figure 5B–D), demonstrating a proof-of-concept of the NK cell effect.

Autologous NK cells alone (Log Rank_10.19_, *p* = 0.0014, 95%CI 0.252–0.523) or when combined with BTZ (Log Rank_5.25_, *p* = 0.0219, 95%CI 0.295–0.408) prolonged animal survival compared to controls (median 98 vs. 68 days and 80 vs. 68 days, respectively, Figure 5D), demonstrating that autologous NK cells were potent in the treatment of GBM cells in vivo. BTZ alone also prolonged animal survival compared to controls (Log Rank_6.57_, *p* = 0.0104, 95%CI 0.284–0.424, median 83 vs. 68 days, Figure 5D). Combination BTZ+NK cells mildly enhanced autologous NK treatment in vivo as 25% (2/8) of animals survived up to 124 days (Figure 5D). Furthermore, mice treated with NK monotherapy showed a tendency to survive longer than those treated with BTZ alone (Log Rank_3.37_, *p* = 0.06, 95%CI 0.245–0.397, median 98 vs. 83 days, respectively, Figure 5D). BTZ can give rise to thrombocytopenia, but blood clotting times and body weight confirmed that the treatments were tolerated (Appendix A).

### 3.11. NK Cell Treatment Attenuates Tumour Proliferation while BTZ Treatment Diminishes Angiogenesis

Tumours from mice under the various treatment protocols displayed histological differences. Vehicle-treated control, BTZ monotherapy, as well as combination BTZ+anti-asialo-GM1+NK-cell-treated BG7 tumours were large, highly cellular and contained areas of pseudopalisading necrosis with pyknotic cells and microvascular proliferations (Figure 5G). NK-cell-treated tumours had reduced Ki67 proliferation indices compared to vehicle-treated controls (*p* < 0.0001) and BTZ+NK-cell-combination-treated tumours (*p* < 0.01, Figure 5E,H). Tumours exposed to BTZ alone (*P* < 0.05) or in combination with NK cells (*p* < 0.01) exhibited diminished angiogenesis compared to NK cell monotherapy, as shown by the diminished CD31 microvascular density (Figure 5F,I). NK-cell-treated tumours contained numerous granulocytic immune cells, some with NK cell morphology. The frequency of granulocytic immune cells was, however, reduced in tumours treated with combination BTZ+NK cells (Figure 5G). End-stage tumours were all confirmed by expression of human-specific nestin (Figure 5J), which revealed a highly invasive growth pattern reminiscent of the parental patient GBM in situ (Figure 5K).

### 3.12. BTZ Penetrates the Blood-Brain Barrier

The ability of BTZ to penetrate the blood–brain barrier has been debated; therefore, we performed intratumour biodistribution analyses and demonstrated that intraperitoneally (i.p.)-administered BTZ did cross the blood–brain barrier and could be detected in greater concentrations in the tumour-bearing mouse brains (42.54 ± 11.86 ng/g tissue) compared to healthy mouse brains with an intact blood–brain barrier (6.06 ± 1.15ng/g tissue; *p* < 0.01, Figure 5L). Furthermore, to confirm that BTZ had penetrated the brain tumour, proteasomal activity of the treated tumour cells was analysed by target inhibition of the 1 and 5 subunit activity in situ. Tumours treated with BTZ, as monotherapy or in combination with NK cells, displayed significantly reduced 1 and 5 subunit activity compared to vehicle control (*P* < 0.0001, both) or NK monotherapy (*P* < 0.01 for BTZ and *P* < 0.001 for combination, Figure 6A), indicating that the drug sufficiently crossed the blood–brain barrier to inhibit the catalytic subunits of the 20S proteasome.

### 3.13. Long-Term Persistence of Adoptively Transferred Human CD56^+^ NK Cells in Vivo

Finally, to investigate the impact of the treatment on the tumour in situ, we dissociated mouse brains and phenotyped the human-specific CD45 negative tumour cell and CD45^+^ immune cell populations (Figure 6B,C). The adoptively transferred human CD56^+^CD3^-^NK cells were detected within the CD45^+^ cell fraction compared to in untreated controls (*p* < 0.05, Figure 6C,D).

In addition, CD45^+^CD56^+^ cells in the brains of animals treated with NK cells significantly expressed higher NKp46 compared to untreated controls (*p* < 0.05, Figure 6D), which confirmed the presence of NK cells. The CD45^+^CD56^+^ gated population in dissociated tumours of NK cell monotherapy and NK+BTZ-combination-treated mice showed significantly increased CD56 MFI when compared to controls after the normalization of data (*p* < 0.05 for both, one-way ANOVA with Turkey’s multiple comparisons test. Moreover, fewer proportions of cells in BTZ-treated tumours expressed HLA-DR, -DP, or -DQ compared to vehicle control, combination BTZ+NK cell (*p* < 0.01 respectively) or BTZ+anti-asialo-GM1+NK-cell-treated BG7-tumour-bearing mice (*p* < 0.001, Figure 6E,G). When NK cell activity was ablated in BTZ+anti-asialo-GM1+NK-cell-treated mice, the levels of class I HLA-A, -B, and -C were significantly reduced compared to combination BTZ+NK cell treatment (*p* < 0.01, Figure 6G). TRAIL-R2 levels were also attenuated in BG7 tumours treated with combination BTZ+NK cells compared to vehicle control or BTZ alone (*p* < 0.05 and *p* < 0.01, respectively, Figure 6F,G).

## 4. Discussion

Immunotherapy is increasingly becoming the fourth pillar of treatment [41] after surgery, radiotherapy and chemotherapy for patients with aggressive solid malignancies. However, immunotherapy is not an effective modality for all malignancies nor for all patients with the same diagnosis. Therapeutic potential of immunotherapy for GBM is not yet unequivocally realised [42,43]. Here, we investigated NK cells as therapeutic effectors for GBM. Allogeneic and autologous NK cells did kill selected tumour cell populations in vitro and the activity was enhanced by combination with BTZ in 3/5 GBM cases. Furthermore, we show potent efficacy of a single dose of autologous NK cells adoptively transferred into the patient-derived GBM xenograft. These adoptively transferred CD56^+^ NKp46^+^ human NK cells persisted to end-stages of survival. The efficacy of NK+BTZ was mediated through increased stress ligands recognized by NK-cell-activating receptors. As proof of concept, we showed that blocking NKG2D receptor–stress ligand interactions induced by BTZ most potently rescued the GBM cell death mediated by NK cells, indicating this pathway as a major mechanism for BTZ sensitization to NK cell lysis. NK cells’ lysis of GBM cells was also executed through extrinsic apoptosis mediated via ligation of TRAIL to cognate TRAIL-R2 (DR5) upregulated on GBM cells after BTZ pretreatment. This is supported by the finding that blocking both TRAIL and TRAIL-R2 rescued GBM cell lysis by approximately 50%. However, in vivo, TRAIL-R2 was downregulated in the remaining end-stage tumour cells, potentially implicating the loss of the receptor in treatment-resistant tumour cells. Previously, BTZ has been demonstrated to sensitize mouse-derived tumour cells to treatment with TRAIL cytokine through induction of DR5 [44,45]. Our study supports these findings as we also observed upregulation of TRAIL-R2 on GBM cells after BTZ pretreatment in vitro.

The data underscore the complexity of NK cell killing mechanisms when augmented by pretreatment of GBM with BTZ. The latter also perturbed mitochondrial bioenergetics and abrogated autophagic flux. Our findings confirm earlier reports that low-dose BTZ increased expression of stress ligands recognised by NKG2D and DNAM-1 NK cell receptors and enhanced cytotoxicity through increased IFNγ production [25,26,44]. Our results further support the investigation of adoptive transfer of NK cells as a treatment for some GBM patients.

NK cells potently secreted IFNγ, perforin and granzyme A in vitro after contact with GBM cells and abridged tumour mitochondrial function by diminishing both ATP production and maximal respiration. Under these conditions, the susceptible cells, ultimately, could not recover to cope with additional energy demands, as indicated by their lower spare respiratory capacity. Granzymes are serine proteinases with substrate specificity for cleavage on aspartate residues [46] where granzyme B induces cytochrome *c* release [47] by cleavage and inactivation of PARP-1 and caspase 3 to generate double-strand DNA breaks. In contrast, granzyme A induces rapid caspase-independent cell death through ssDNA breaks [48,49]. Thus, tumour cells that are resistant to caspase-mediated cell death, including GBMs that overexpress Bcl-2, might be more sensitive to granzyme A. IFNγ has also previously been implicated in the modulation of levels of HLA ligands [50]. NK- and T-cell-derived IFNγ levels in GBM patients’ tumours may be low due to limited NK cell infiltration into the tumour parenchyma [51].

Indeed, when NK cell activity was ablated in vivo, the levels of class I HLA-ABC were attenuated, as were HLA-DP, -DQ, -DR levels after BTZ treatment. These findings are consistent with the fact that the 26S proteasome is necessary for the processing of both class I HLA and HLA-DP, -DQ, -DR [52,53] and subsequent antigen presentation; however, we did not investigate the implications for antigen presentation. This was largely due to the fact that our choice of in vivo model limited the potential to investigate the effects of the host immunity that could have further potentiated the impact of NK cells.

Due to its molecular weight, the degree to which BTZ actively penetrates the blood–brain barrier (BBB) is frequently debated. However, several studies detected the drug in brain tumours in both patients and mice [54,55] after intravenous injection. An increased capacity to cross over to the brain tissue may be assisted by the natural disruption of the BBB during malignant progression [55]. Our LC–MS/MS results confirmed that BTZ crossed the BBB and was bioactive at the orthotopic site since tumours from BTZ-treated animals displayed diminished angiogenesis and attenuated β1, β5 chymotryptic-like activity at levels that were sufficient to improve survival. Furthermore, we found reduced tumour volumes in BTZ- and BTZ+GM1-NK-treated mice compared to vehicle-treated control mice. In vivo, BTZ alone did not mirror the clear effect shown in vitro. Although tumour growth was reduced in BTZ-treated animals relative to vehicle control, animals treated with BTZ did not live longer than those treated with NK cell monotherapy, indicating inferior single agent efficacy. This result could explain why combination with NK cells did not result in a substantial benefit when compared to NK cells alone. BTZ potentially reduced NK cell viability as observed in vitro, and minimised the effectiveness of the surviving NK cells. Our findings thus emphasize the need for careful timing of the combination to ensure that BTZ does not kill off the transferred NK cells.

Recently, BTZ was shown to synergise with oncolytic herpes simplex virus through induction of necroptotic tumour death that enhanced NK cell cytotoxicity [56]. Here, we also demonstrated that BTZ alone or in combination with NK cells inhibited autophagic flux, as evidenced by the accumulation of p62 and a significantly increased ratio of autophagosomes to degradative autolysomes in vitro. Autophagy has a dual role in promoting tumour cell survival and death [57,58] at different stages of disease or treatment [59,60], and in a cell-specific manner in response to proteasome inhibitors. On one hand, cancer cells increase autophagic flux to degrade damaged and misfolded proteins (which more frequently occurs in malignant cells as a result of mutations) as a means to reprogram their metabolic needs under nutrient-limiting conditions, due to hypoxia or unstable angiogenesis [58]. On the other hand, studies show that disrupted autophagic flux increases oxidative stress, potentially leading to double-stranded DNA breaks and gene amplifications [29,61,62]. Glioblastoma cells also increase the autophagic flux in response to genotoxic stress and organelle damage induced by chemotherapy [60,63,64]. GBM cells may instigate flux as a means to terminate gene transcription since transcription factors are short-lived proteins that are typically degraded by the 26S proteasome or activated through targeted degradation of suppressor subunits, exemplified in a canonical NFκB signalling cascade. By these means, autophagic flux enables the cells to maintain or lose genomic integrity, which impacts the cellular processes that define the hallmarks of cancer [58].

Nevertheless, the impact/contribution of abrogated autophagic flux to the efficacy of BTZ+NK cell treatment is difficult to ascertain unilaterally. Using a lysotracker reagent and LC3-puncta, we showed the accumulation of lysosomes in GBM treated with both BTZ alone and combination BTZ+NK cells, which also showed increased yellow puncta/autophagosomes, indicating abrogated autophagic flux. On the other hand, in NK cell monotherapy, negligent LC3-puncta were observed and red puncta/autophagosomes were observed, consistent with sustained autophagic flux. In addition, increased % or MFI of p62 in GBM cells was observed after BTZ monotherapy. In vivo, NK cell inhibition attenuated p62 levels compared to BTZ- and combination-treated tumours. Overall, our results indicate that BTZ inhibits the autophagic flux, while NK cells sustain the autophagic flux. Combination teatment perturbed GBM mitochondrial function and induced PARP-1 and caspase-3-dependent apoptosis. Therefore, the autophagy disruption observed in BTZ+NK combination therapy is caused by BTZ and killed cells in a caspase-independent death mechanism. These findings may point to efficacy of the combination strategy mediated by two independent mechanisms, explaining the 25% prolonged survival at the tail of the survival curve. Thus, accumulated autophagosomes and p62 bodies in treatment-responsive BG7 cells in vitro and in vivo, as well as in P3 GBM cells compared to treatment-resistant 2012-018 GBM cells might support a role in accentuating tumour cell death.

## 5. Conclusions

Combination immunotherapy may enhance the therapeutic efficacy in solid tumours that are characterized by high molecular heterogeneity and are difficult to treat. Pretreatment of GBM in vitro with BTZ for 24 h induced stress ligand expression, disrupted their autophagic flux and enhanced the killing efficacy of NK cells in 3/5 cases. In vivo, autologous NK cell monotherapy and combination with BTZ significantly reduced tumour volumes and prolonged animal survival. A total of 25% of GBM-bearing mice experienced prolonged survival with combination therapy compared to monotherapy. This modest effect on survival might be explained by the BTZ effect on NK cell viability shown in vitro. Therefore, further studies are needed to address dose and schedule timing. Our study also highlights the need for investigating the potential for augmenting responses via bystander immunological effects.

## Figures and Tables

**Figure 1 cancers-11-00996-f001:**
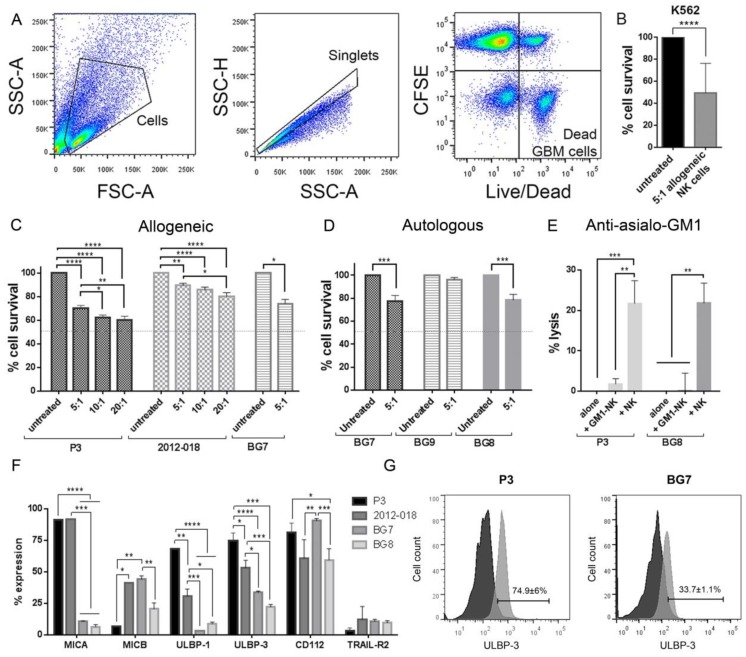
The effect of treatment with natural killer (NK) cells against glioblastoma (GBM). (**A**) A gating strategy for cytotoxicity assays showing analysis from left to right of cells, singlets, separation of effector (NK cells) and target (GBM cells) cells, and gating on live cells. The % fraction of surviving cells determined from flow cytometry analyses in (**B**) K562 cells treated with 5:1 of allogeneic NK cells; (**C**) P3, 2012-018 and BG7 GBM cells treated with 5:1, 10:1 and 20:1 of allogeneic NK cells; and (**D**) BG7, BG9 and BG8 GBM cells treated with autologous 5:1 of NK cells. (**E**) % cell lysis, as determined from flow cytometry, of P3 and BG8 GBM cells after allogeneic 5:1 NK cells alone and 5:1 NK cells inhibited with anti-asialo-GM1. (**F**) The % expression, determined with flow cytometry, of NKG2D ligands, CD112 and TRAIL-R2 in untreated P3, 2012-018, BG7 and BG8 cells. (**G**) ULBP3 histograms in P3 (left) and BG7 (right) cells (control Fluorescent Minus One (FMOs) dark grey and stained sample light grey). (**H**) Dot plot showing (from left to right) CD56 vs. KIR2DL1, CD56 vs. KIR3DL2/1 and CD57 vs. KIR3DL2/1 in representative NK cells. The % expression of (**I**) killer immunoglobulin-like receptors (KIR) receptors and (**J**) NK cell markers on donor-derived-NK cells. (**K**) ELISA to determine concentration in pg/mL of IFNγ TNF perforin and granzymes A and B in supernatant of P3 and 2012-018 cells treated with allogeneic NK cells and NK cells alone. (**L**) Profile from the Seahorse assay for mitochondrial respiration based on the oxygen consumption rate (OCR) and sequential drug injection for P3 and 2012-018 GBM cells alone and treated with allogeneic NK cells. (**M**) Graph showing measurements of maximal respiration, ATP production and spare respiratory capacity of P3 and 2012-018 cells alone (black) and treated with allogeneic NK cells (grey). Data represents the mean ± SEM of *n* = 3 independent experiments. Two-way ANOVA, Bonferroni’s multiple comparison test, * *P* < 0.05; ** *p =* 0.01; *** *P* < 0.001; **** *P* < 0.0001.

**Figure 2 cancers-11-00996-f002:**
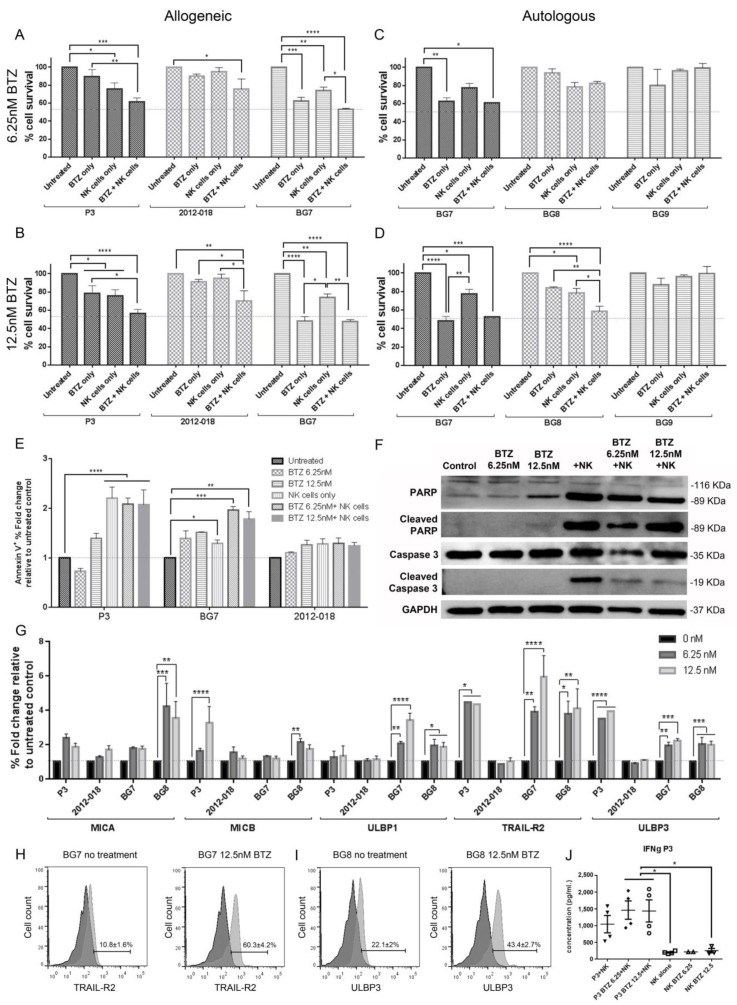
Combination treatment of bortezomib and NK cells against GBM cells. The % fraction of surviving P3, 2012-018 and BG7 cells treated with bortezomib (BTZ), with 5:1 of NK cells or in combination BTZ and 5:1 of NK cells; treated with allogeneic NK cells and (**A**) 6.25 nM or with (**B**) 12.5 nM BTZ, or treated with autologous NK cells and (**C**) 6.25 nM or (**D**) 12.5 nM BTZ. (**E**) The % fold change relative to the untreated control of early apoptotic (Annexin V^+^) P3, BG7 and 2012-018 GBM cells after treatment with 6.25 nM BTZ, 12.5 nM BTZ, 5:1 NK cells and combination of 6.25 nM or 12.5 nM BTZ with 5:1 NK cells. (**F**) Western blot analysis of poly (ADP-ribose) polymerase (PARP) and caspase 3 proteins in lysates (20 µg of protein) from P3 cells untreated or treated with 6.25 nM and 12.5 nM BTZ, 5:1 NK cells or combination 6.25 nM and 12.5 nM of BTZ+5:1 NK cells. Glycerol-3-phosphate dehydrogenase (GAPDH) was used as a loading control and for the normalization of densitometry measurements. (**G**) The % fold change relative to untreated control cells (black) in P3, 2012-018, BG7 and BG8 of NKG2D ligands: MICA, MICB, ULBP1, ULBP3, and the receptor TRAIL-R2 after treatment with 6.25 nM (dark grey) and 12.5 nM (light grey) of BTZ. Histograms showing expression of (**H**) TRAIL-R2 in BG7 and (**I**) ULBP3 in BG8 cells without treatment and after 12.5 nM BTZ for 24 h. (**J**) Concentration in pg/mL of IFNγ, from ELISA, in supernatants of P3 GBM cells under the different treatment conditions. Data represents the mean ± SEM of *n* = 3 independent experiments. Two-way ANOVA, Bonferroni’s multiple comparison test, * *P* < 0.05; ** *P* < 0.01; *** *P* < 0.001; **** *P* < 0.0001. Lysates for proteins shown in Western blots in Figure 2F and Figure 3A are from the same experiment, and blots represent two independent experiments.

**Figure 3 cancers-11-00996-f003:**
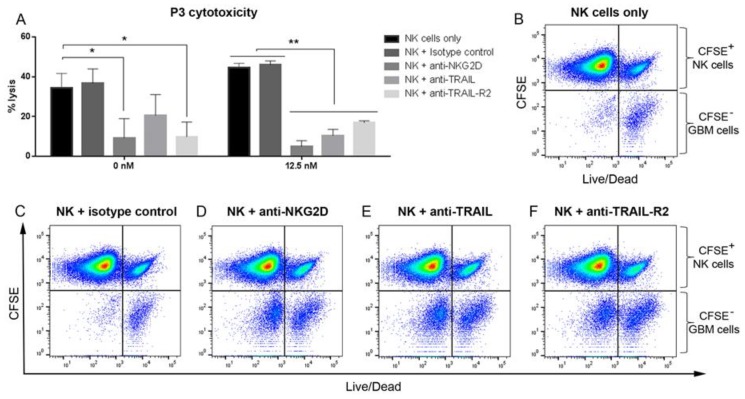
Blocking NKG2D, TRAIL and TRAIL-R2 rescues GBM cells from NK lysis. (**A**) The % of P3 GBM lysis by NK cells before and after treatment with 12.5 nM of BTZ, in the presence or absence of blocking antibodies for NKG2D, TRAIL and TRAIL-R2, and isotype control. Two-way ANOVA, Bonferroni’s multiple comparison test, * *P* < 0.05; ** *P* < 0.01. Dot plots show CFSE-labelled NK cells and unlabelled P3 cells vs. Live/Dead positive P3 cells pretreated with 12.5 nM BTZ prior to treatment with (**B**) NK cells only, and NK cells in the presence of (**C**) isotype control, (**D**) anti-NKG2D, (**E**) anti-TRAIL, and (**F**) anti-TRAIL-R2. Data represents the mean ± SEM of *n* = 3 independent experiments.

**Figure 4 cancers-11-00996-f004:**
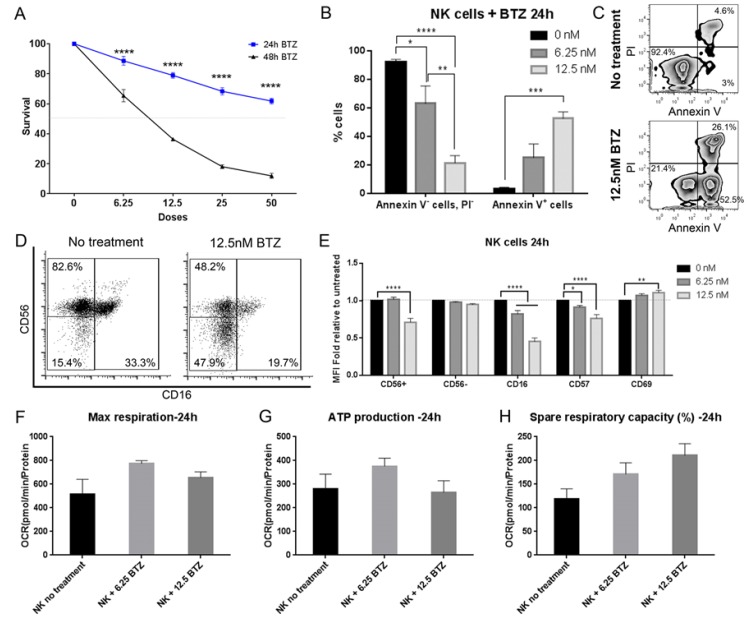
Bortezomib modifies NK cells to mature activated phenotypes. (**A**) Survival curves of NK cells treated with 0–50 nM of BTZ for 24 h (blue) and 48 h (black). (**B**) The % of early apoptotic (Annexin V^+^) and healthy (Annexin V^−^, PI^−^) NK cells after treatment with 6.25 nM (dark grey) and 12.5 nM (light grey) of BTZ, determined using flow cytometry. (**C**) Dot plots showing apoptosis gating strategy using PI vs. Annexin V with no treatment (top) and with 12.5 nM BTZ (bottom). (**D**) CD56 vs. CD16 dot plots of NK cells untreated and treated with 12.5 nM BTZ. (**E**) MFI fold relative to untreated NK cells of NK cell markers after treatment with 6.25 nM (dark grey) and 12.5 nM (light grey) BTZ. Measurements of NK cells for (**F**) maximal respiration, (**G**) ATP production and (**H**) spare respiratory capacity after the treatments. Data represents the mean ± SEM of *n* = 3 independent experiments. Two-way ANOVA, Bonferroni’s multiple comparison test, * *P* < 0.05; ** *P* < 0.01; *** *P* < 0.001; **** *P* < 0.0001.

**Figure 5 cancers-11-00996-f005:**
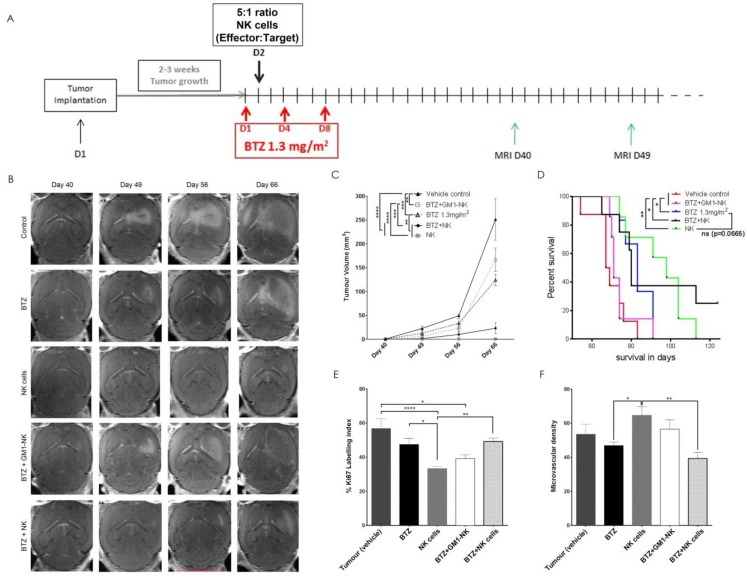
Treatment of GBM-bearing mice with bortezomib+NK cells. (**A**) Tumour implantation (~10^5^ BG7 cells), treatment schedule (1.3 mg/m^2^ BTZ at days 1, 4 and 8 and, 5 × 10^5^ autologous NK cells 24 h after the first dose of BTZ) and frequency of MRI scans (7–10 days). (**B**) Axial T1-weighted with contrast longitudinal MRI follow-up of the same animal from each treatment group and follow-up post-implantation on days 40, 49, 56 and 66. (**C**) The mean ± SEM of tumour volumes (mm^3^) and (**D**) Kaplan–Meier curves showing the % survival in days of all animals in the indicated treatment groups. The % mean ± SEM of (**E**) Ki67 labelling index and (**F**) CD31 microvascular density in all animals in the indicated treatment groups. Data represents *n* = 3 independent experiments. Two-way ANOVA, Bonferroni’s multiple comparison test, * *P* < 0.05; ** *P* < 0.01; *** *P* < 0.001; **** *P* < 0.0001. (**G**) Hematoxylin and Eosin (H&E) staining of sections from tumours treated with NK cells or BTZ+NK cells (magnification 200× *g*), (**H**) Ki67 staining (brown) hematoxylin counterstain, (**I**) CD31 staining (brown) for tumour microvessels, (**J**) human-specific nestin staining (brown) and (**K**) whole brain image of nestin staining (brown) of tumour tissue from one representative animal from each of the treatment groups indicated. (**G**–**J**) Magnification 400× *g*, scale bar 100 μM, arrows: NK cells, (**K**) magnification 10×, scale bar 1 mm, animals sacrificed on days 67, 74, 98, 74 and 65, from left to right, respectively. (**L**) Table showing Mean ± S.E.M of BTZ tissue concentrations (ng/g) in tumour-bearing mouse brains vs. non-tumour bearing controls with an intact blood–brain barrier (top left panel). Typical chromatograms in positive multiple reaction monitoring (MRM) mode for bortezomib in vehicle control mouse brain tissue (top right panel), vehicle control mouse brain tissue spiked with the BTZ analyte (bottom left panel) and mouse brain tissue treated with BTZ 1.3 mg/m^2^ on days 1, 4, 8 and 11 (bottom left panel).

**Figure 6 cancers-11-00996-f006:**
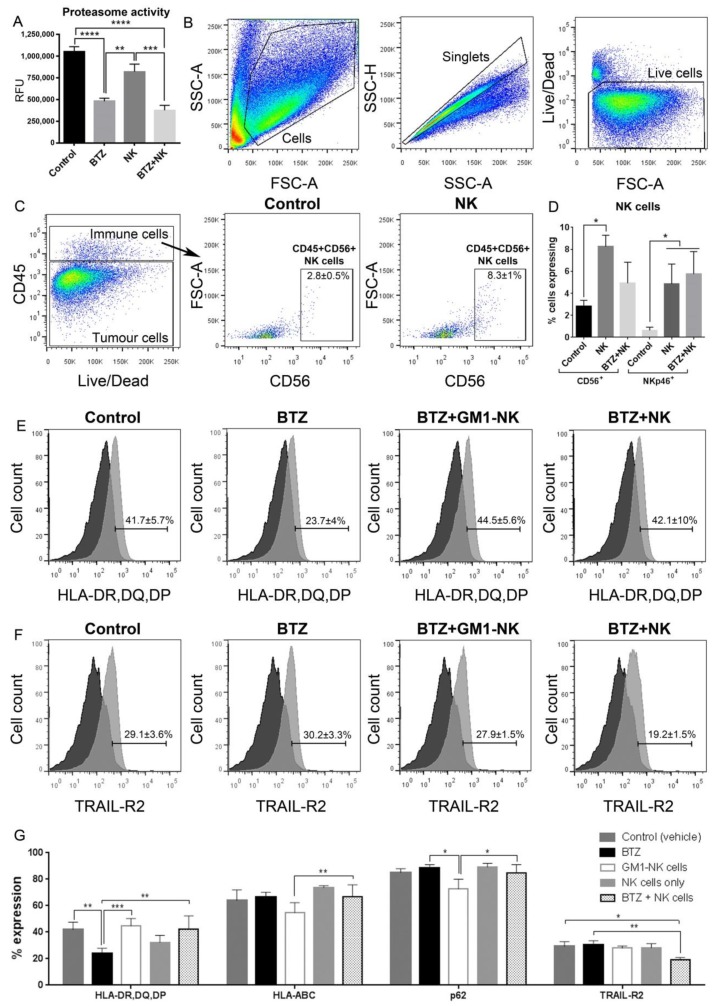
Marker expression of the in vivo experiment. (**A**) Relative fluorescence units (RFU), as a measure of 20S proteasome activity, in dissociated brains from mice from each of the treatment groups indicated. Flow cytometry gating strategy of dissociated tumour-bearing mouse brains (from left to right): (**B**) cells, singlets and live cells and (**C**) CD45^-^ tumour cells/CD45^+^ immune cells, CD45^+^CD56^+^ NK cells in control and NK-cell-treated mice. Bar graph showing (**D**) % of CD45^+^CD56^+^ and CD45^+^CD56^+^NKp46^+^ NK cells and One-way ANOVA, Bonferroni’s multiple comparison test. Histograms showing (**E**) HLA-DR, -DQ, -DP and (**F**) TRAIL-R2 expression in tumour cells from control, BTZ-treated, BTZ+GM1-treated NK cells and mice treated with BTZ+NK cells (control FMOs black and stained sample grey). (**G**) % of cells expressing HLA-DR, -DQ, -DP, HLA-ABC, p62 and TRAIL-R2 in dissociated brains from mice from each of the treatment groups indicated. Brains used in flow cytometry were from animals sacrificed in the following range of days after starting of experiment: 67–74 days Control, 73–83 days BTZ, 74–104 days NK cells, 70–74 days BTZ+GM1-NK and 65–80 days BTZ+NK cells. Data represents the mean ± SEM of *n* = 3–6 dissociated brains/treatment group. Two-way ANOVA, Bonferroni’s multiple comparison test, * *P* < 0.05; ** *P* < 0.01; *** *P* < 0.001; **** *P* < 0.0001.

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
