# Peer review of "Pretreatment of Glioblastoma with Bortezomib Potentiates Natural Killer Cell Cytotoxicity through TRAIL/DR5 Mediated Apoptosis and Prolongs Animal Survival"

_cancers, 2019, doi:10.3390/cancers11070996_

Reviewer 1 Report

The authors herein examined whether pretreatment of glioblastoma (GBM) with the proteasome inhibitor bortezomib (BTZ) might sensitize tumor cells to NK cell lysis by inducing stress antigens recognized by NK activating receptors. They demonstrated that NK cells alone or in combination with BTZ inhibit tumor growth, but at the same time scheduling of BTZ in vivo requires further investigation to maximize its contribution to efficacy of the combination regimen. The findings would be interesting and useful for the future application of immunotherapies against the deadly brain tumor GBM, but several revisions are necessary to tighten the manuscript and further clarify their argument.

1. Introduction section is rather lengthy, and it should be much more concise to convey their hypothesis and intent.

2. They should explain in more detail why some GBM cells showed resistance to NK cell-mediated lyses (e.g. BG9 tumor cells in Figure 1).

3. Please briefly explain the difference in granzyme A and B, and their expression in NK cells in the study.

4. To look at ECAR with Seahorse in the same setting would complement the OCR data in the present study.

5. It would be preferable if they could differentiate the effect of combination treatment on apoptosis via intrinsic or extrinsic pathways (Figure 2).

6. Could the treatment with BTZ possibly enhance the somatic mutation and neoantigen burden in glioma cells, which could affect the efficacy of immunotherapies? 

7. Also, please briefly discuss if the genotype of gliomas (e.g. IDH status) could affect the efficacy of NK cell treatment.

8. In addition to flowcytometry (Figure 6), it would be interesting to immunohistochemically look at in situ expression of NK cell markers including cytotoxic ones and stress-related molecules in GBM-bearing mice models (Figure 5).

9. Discussion section is rather fragmented, which should be more tightened.

Author Response

Reviewer 1

The authors herein examined whether pretreatment of glioblastoma (GBM) with the proteasome inhibitor bortezomib (BTZ) might sensitize tumor cells to NK cell lysis by inducing stress antigens recognized by NK activating receptors. They demonstrated that NK cells alone or in combination with BTZ inhibit tumor growth, but at the same time scheduling of BTZ in vivo requires further investigation to maximize its contribution to efficacy of the combination regimen. The findings would be interesting and useful for the future application of immunotherapies against the deadly brain tumor GBM, but several revisions are necessary to tighten the manuscript and further clarify their argument.

1. Introduction section is rather lengthy, and it should be much more concise to convey their hypothesis and intent.

We agree with the reviewer and we have shortened, condensed and modified the introduction so our hypothesis gets across.

2. They should explain in more detail why some GBM cells showed resistance to NK cell-mediated lyses (e.g. BG9 tumor cells in Figure 1).

In supplementary table IV we show that BG9 patient’s NK cells possessed inhibitory KIR2DL2 and KIR2DL3 educated within the cognate ligand HLA-C1 *07:02. This would render these NK cells inhibitory to self-cells. BG8 and BG7 tumours that were responsive to autologous NK cells did not possess HLA-C1 *07:02 epitope. As this was a distinguishing feature of BG9 tumour, we speculate this might be the reason for the resistance to autologous NK cells. The resistance could also be explained by low basal NKG2D ligands expression and/or inability of BTZ to enhance their expression as we observed in 2012-018. Unfortunately, stress ligand expression in BG9 cells could not be assessed by flow due to unavailability of the cells for large scale analysis.

3. Please briefly explain the difference in granzyme A and B, and their expression in NK cells in the study.

We address this suggestion in the discussion. “Granzymes are serine proteinases with substrate specificity for cleavage on aspartate residues [42] where granzyme B induces cytochrome c release [43] by cleavage and inactivation of PARP-1 and caspase 3 to generate double strand DNA breaks. In contrast, Granzyme A induces rapid caspase-independent cell death through ssDNA breaks [44,45]. Thus, tumour cells that are resistant to caspase-mediated cell death, including GBMs that overexpress Bcl-2, might be more sensitive to granzyme A.” (lines 615-620)

4. To look at ECAR with Seahorse in the same setting would complement the OCR data in the present study.

We agree that ECAR information is a good complement of OCR when investigating cell energetics. Oxygen consumption rate (OCR) gives information of the mitochondrial respiration, thus it provides insight into mitochondrial function. Extracellular acidification rate (ECAR) gives information about glycolytic function of the cell, and it is normally used to assess the energy metabolism. Despite ECAR measurements would have provided us insights on the energy metabolism of the cells, our goal was to investigate whether the mitochondria function was disrupted on not with the treatment. Therefore, we only performed Seahorse experiments measuring OCR.

5. It would be preferable if they could differentiate the effect of combination treatment on apoptosis via intrinsic or extrinsic pathways (Figure 2).

We have addressed this in the section 3.5, 3.6 and 3.7.

6. Could the treatment with BTZ possibly enhance the somatic mutation and neoantigen burden in glioma cells, which could affect the efficacy of immunotherapies? 

Bortezomib has previously been shown to induce immunogenic death (Bezu et al, 2015) where stress responses resulting in emission of danger associated molecular patterns (DAMPs) and high mobility group protein -1 (HMGP1) from the dying cells as well as exposure to endoplasmic reticulum and calreticulin on the cell surface, ATP production and type I interferon could stimulate innate and adaptive cells. Although it is possible that BTZ treatment could enhance somatic mutations, this has previously not been specifically addressed. Determining neoantigen burden in GBM is an extensive process that is regrettably beyond the scope of the current work.

7. Also, please briefly discuss if the genotype of gliomas (e.g. IDH status) could affect the efficacy of NK cell treatment.

We verified IDH status of the GBMs used in the study through diagnostic molecular pathology performed neuropathologist prof. Hrvoje Miletic (now included as co-author). 4/5 of the GBMs were IDH wildtype while status for 2012-018 was not available.

8. In addition to flow cytometry (Figure 6), it would be interesting to immunohistochemically look at in situ expression of NK cell markers including cytotoxic ones and stress-related molecules in GBM-bearing mice models (Figure 5).

Unfortunately, our available antibodies for NK cells are not optimized for paraffin-embedded tissues and do not show specificity for target when single marker staining is used. Therefore, we performed new staining for NKp46 (NK cell marker) of dissociated brains to confirm our results detecting NK cells based on CD45+CD56+CD3- phenotype. We could detect NKp46+ NK cells within the CD45+CD56+CD3- phenotype, confirming the presence of  NK cells in treated brains at end-stage (data added in Figure 6D, lines 553-555).

9. Discussion section is rather fragmented, which should be more tightened.

We have modified and shortened the discussion according to reviewer’s suggestion.

Reviewer 2 Report

It has been analyzed the antitumor effect of bortezomib and NK cells combination  on some glioblastoma  (GBM) cell lines in vitro. In addition, this therapeutic combination is more effective than bortezomib or NK cells alone in reducing GBM growth in vivo.
This work is well written and clearly understandable.Methods are adquate, data are well presented and the conclusion are in line with the results.
There are some points that should be addressed.
1. The induction of CD69 on bortezomib treated NK cells should be shown as FACS data on NK cells gated as AV and PI negative cells. Indeed, it seems that the induction of this receptor is not really strong and it is clear from data shown that some NK cells are dying when incubated with bortezomib. Thus, it is relevant to show that only living cells can upregulate CD69 at the cell surface.
2. The finding that freshly isolated NK cells are sensitive to bortezomib would indicate that  effector cells are killed by bortezomib and thus the increment of GBM cell killing can be due to the damage induced on NK cell by this drug. It is important to analyze that during cytotoxic experiments with GBM cells the effect of bortezomib on NK cells and to show whether bortezomib can influence the vitality of activated NK cells.

Author Response

Reviewer 2

It has been analyzed the antitumor effect of bortezomib and NK cells combination on some glioblastoma (GBM) cell lines in vitro. In addition, this therapeutic combination is more effective than bortezomib or NK cells alone in reducing GBM growth in vivo.
This work is well written and clearly understandable. Methods are adequate, data are well presented and the conclusions are in line with the results.
There are some points that should be addressed.

1. The induction of CD69 on bortezomib treated NK cells should be shown as FACS data on NK cells gated as AV and PI negative cells. Indeed, it seems that the induction of this receptor is not really strong and it is clear from data shown that some NK cells are dying when incubated with bortezomib. Thus, it is relevant to show that only living cells can upregulate CD69 at the cell surface.

In FACS data, Live/Dead Fixable Near-IR Dead Cell Stain Kit (Invitrogen) was used to gate out dead cells and to make sure that in all measurements, analyses were performed in live cells, using the same gating strategy showed in Figure 6 A-B (in manuscript). As shown by example gating, within the 98.6% live NK cells, 36.5% were CD69 positive. This is important as dead cells bind unspecifically to antibodies.

(Please see figure in attached file)

From left to right: Dot plots showing SSC-A vs. FSC.A (cells); SSC-H vs. SSC-A (single cells) and Live/Dead vs. FSC-A (live cells); and histogram showing Count vs. CD69 in PE, dark grey is CD69 FMO and light grey the stained NK cell sample.

2. The finding that freshly isolated NK cells are sensitive to bortezomib would indicate that effector cells are killed by bortezomib and thus the increment of GBM cell killing can be due to the damage induced on NK cell by this drug. It is important to analyze that during cytotoxic experiments with GBM cells the effect of bortezomib on NK cells and to show whether bortezomib can influence the vitality of activated NK cells.

All experiments involving NK cells were performed using NK cells activated and expanded for 14 days in IL-2 (500U/ml) and IL-15 (5ng/ml). In vitro, vitality of activated NK cells was affected by bortezomib, most prominently after 48h. In the combination treatment in vitro, NK cells were always added to bortezomib pretreated GBM cells after removal of drug containing medium. However, in vivo we used anti-asialo-GM1 ganglioside to deplete NK cells in order to verify the impact of NK cells to the combination treatment. Unfortunately, the effect of bortezomib on NK cells in vivo was not directly addressed since it would require sacrificing of animals at set time points and would compromise the survival analyses. We had 5 treatment groups each with 7-8 animals. Sacrificing several animals per group at set time points, in addition to survival analyses would have been a challenging argument for the 3Rs of ethics of animal experiments to ethics board.

Reviewer 3 Report

The issue is interesting and authors present a great amount of data, but the presentation is confused and too long.

English language must be revised.

Title: 

The title of the manuscript does not synthetize exhaustively the content of the paper.

Abstract, Methods:

In this paragraph, authors must shortly summarize their in vitro and in vivo tests.

Introduction: 

This section is too long and confused. The rationale is unclear. Lines 92-98 anticipate results and must be removed.

Results:

The last part, regarding in vivo experiments, must be better explained and clarified (paragraphs 3.10, 3.11, 3.12)

Discussion:

Also this section is too long and confused.

Conclusions:

This paragraph should summarize the principal data emerging from the study and suggest potential future research.

Author Response

Reviewer 3

The issue is interesting and authors present a great amount of data, but the presentation is confused and too long.

Thank you, we have shortened the introduction and discussion.

English language must be revised.

We have sent the revised manuscript for professional English editing.

Title: The title of the manuscript does not synthetize exhaustively the content of the paper.

We have modified the title according to reviewer’s suggestions: “Pretreatment of glioblastoma with bortezomib potentiates Natural Killer cell cytotoxicity through TRAIL/DR5 mediated apoptosis and prolongs animal survival”

Abstract, Methods: In this paragraph, authors must shortly summarize their in vitro and in vivo tests.

We have modified the paragraph, lines 22-28.

Introduction: This section is too long and confused. The rationale is unclear. Lines 92-98 anticipate results and must be removed.

Thank you. We have shortened the introduction to one page, modified the text and specified the hypothesis to make the rational clearer. We have removed the sentence anticipating the results.

Results: The last part, regarding in vivo experiments, must be better explained and clarified (paragraphs 3.10, 3.11, 3.12)

We have revised and synthesized paragraphs 3.10, 3.11, 3.12 and added a new section (3.12) that more precisely describes BTZ penetration to the blood brain barrier.

Discussion: Also this section is too long and confused.

Thank you. We have revised the discussion section condensing it to two pages.

Conclusions: This paragraph should summarize the principal data emerging from the study and suggest potential future research.

We have modified the conclusions.

Round  2

Reviewer 1 Report

The reviewer would like to thank the authors for carefully and sufficiently addressing all the questions and comments raised here. The manuscript is now ready for acceptance without any further revision.

Author Response

We thank the reviewer for the thorough review of our work

Reviewer 3 Report

Authors have modified the paper as suggested by reviewers.

I only suggest:

1) Introduction section, the rationale of the association bortezomib-NK cells should be better explained;

2) Introduction section, please remove the last 3 lines (79-81);

3) there are still some grammar mistakes

Author Response

Thank you for the thorough review of our work.

1) Introduction section, the rationale of the association bortezomib-NK cells should be better explained;

We have now included some sentences in the introduction that provide the rational for combining bortezomib with NK cells

2) Introduction section, please remove the last 3 lines (79-81);

We have deleted these lines

3) there are still some grammar mistakes

We have carefully read through the manuscript and corrected any residual grammatical errors. The paper was proof read by MDPI´s own language editors.